# PROTOTYPE MATCHING NETWORKS FOR LARGE-SCALE MULTI-LABEL CLASSIFICATION

**Jack Lanchantin, Arshdeep Sekhon, Ritambhara Singh, & Yanjun Qi**
University of Virginia, Department of Computer Science
{jjl5sw, as5cu, rs3zz, yanjun}@virginia.edu

## ABSTRACT

One of the fundamental tasks in understanding genomics is the problem of predicting Transcription Factor Binding Sites (TFBSs). With more than hundreds of Transcription Factors (TFs) as labels, genomic-sequence based TFBS prediction is a challenging multi-label classification task. There are two major biological mechanisms for TF binding: (1) sequence-specific binding patterns on genomes known as "motifs" and (2) interactions among TFs known as "co-binding effects". In this paper, we propose a novel deep architecture, the Prototype Matching Network (PMN) to mimic TF binding mechanisms. Our PMN model automatically extracts prototypes for each TF through a novel prototype-matching loss. We use the notion of a set of prototypes and an LSTM to learn how TFs interact and bind to genomic sequences. On a TFBS dataset with 2.1 million genomic sequences, the PMN significantly outperforms baselines and validates our design choices empirically. The proposed architecture is accurate, and also models the underlying biology.

## 1 INTRODUCTION

To understand genomics, and in turn diseases such as cancer, predicting and understanding Transcription Factor Binding Sites (TFBSs) is essential. Transcription Factors (TFs) are proteins which bind (i.e., attach) to DNA and control whether a gene is expressed or not. TFs are known to bind to sequence-specific patterns on genomes, known as "motifs". If a TF binds in the absence of its motif, or it does not bind in the presence of its motif, then it is likely there are external causes such as interactions with other TFs, known as co-binding effects (Wang et al., 2012). Thus, when designing a genomic-sequence based TFBS predictor, we should consider both: (1) how to automatically extract "motif"-like features and (2) how to model the co-binding patterns and consider such patterns in predicting TFBSs. We address both by proposing the prototype matching network (PMN).

To extract motif-like features, we implement a CNN encoder, as commonly used in TFBS prediction. To model the dependencies among output labels, we introduce a combinationLSTM, which updates the encoder representation of the input conditioned on the output labels. The combinationLSTM models how the embedding of a test sample matches to a combination of relevant prototypes. Using multiple "hops", the combinationLSTM updates the embedding of the input sequence by searching for which TFs (prototypes) are relevant in the label combinations. Instead of explicitly modeling interactions among labels, we try to use the combinationLSTM to mimic the underlying biology. The combinationLSTM tries to learn prototype embedding and represent high-order label interactions through a weighted sum of prototype embeddings.

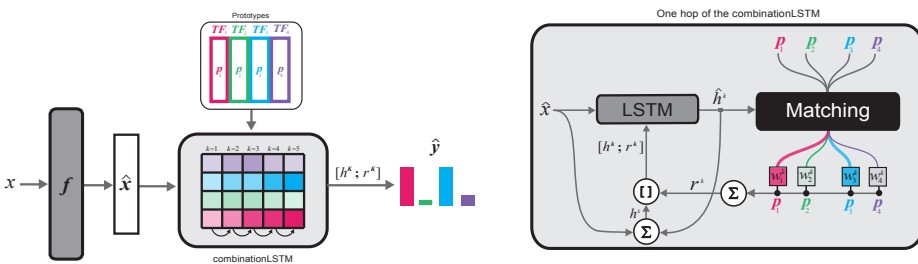

Figure 1: **Prototype Matching Network Model**. Left: overview. Right: combinationLSTM internals

## 2  PROTOTYPE MATCHING NETWORKS

**Model Overview** (Figure 1) Given DNA sequence $x$ (composed of letters A,C,G,T) of length $T$, we want to classify $x$ as a positive or negative binding site for each transcription factor $TF_1, ..., TF_\ell$ (i.e. multi-label binary classification). To do this, we seek to update $x$ by matching it to a set of $\ell$ learned TF prototype vectors, $\{p_1, ..., p_\ell\}$, where each prototype loosely represents the TF label. Since TFs may bind or not bind based on other TFs, we model the interactions among TFs before prediction.

**Embedding The Sequence and Prototypes** The input sequence $x \in \mathbb{R}^{4 \times t}$ is encoded using a function $f$ (3-layer CNN, which has shown to be sufficient for genomic feature extraction) to produce sequence embedding $\hat{x} \in \mathbb{R}^d$. i.e. $\hat{x} = f(x)$. The prototypes are produced by a multiplication of the identity matrix I and a learned lookup table matrix $W \in \mathbb{R}^{|TFs| \times d}$. $P = IW$, where each vector $p_i$ in matrix $P$ is a prototype. Prototypes are forced to correspond to specific TFs using a prototype loss.

**LSTM to learn Label Interactions and Update the Sequence Embedding** The main idea is that we want to update the sequence embedding $\hat{x}$ conditioned on matching against the prototypes. Since interactions among TFs influence binding, we cannot simply match the sequence to the prototypes. To obtain TF interactions, we use a combinationLSTM, similar to the attention LSTM in Vinyals et al. (2016). The combinationLSTM uses $K$ "hops" to process the prototypes $p_1, p_2, ..., p_\ell$ by matching against an updated sequence embedding $\hat{h}^k$. The hops allow the combinationLSTM to update the output vector based on which TFs match simultaneously. At each hop, the LSTM accepts a constant $\hat{x}$, a concatenation of the previous LSTM hidden state $h^{k-1}$ and read vector $r^{k-1}$, as well as the previous LSTM cell output $c^{k-1}$. $r^0$ is initialized with the mean of all prototype vectors, $\frac{1}{|p|} \sum_i^{|p|} p_i$.

The output hidden state $\hat{h}^k$ is matched against each prototype using cosine similarity. Since this similarity is in the range [-1,1], we feed this output through a sigmoid function to produce the similarity score $w_i^k$ at hop $k$ in (0,1). The read vector $r^k$ is updated by a weighted sum of the prototype vectors using the matching scores. At each hop, $h^k$ is updated using the current LSTM output hidden state and sequence embedding:

$$\hat{h}^k, c^k = \text{LSTM}(\hat{x}, [h^{k-1}; r^{k-1}], c^{k-1}) \tag{1}$$

$$r^k = \sum_{i=1}^{|p|} w_i^k p_i \tag{2}$$

$$w_i^k = 1 / \left(1 + e^{-\epsilon c(\hat{h}^k, p_i)}\right) \tag{3}$$

where $c(u, v)$ is cosine similarity, and $\epsilon$ is a hyperparameter of the sigmoid function that pushes the similarity output closer to 0 or 1 (we use $\epsilon$=20). In the TFBS task, eq. 2 is the important factor for modelling TF combinations because $r^k$ can model multiple prototypes matching at once through a linear combination. Furthermore, $K$ hops is needed because a certain TF binding may influence other TFs in a sequential manner. I.e, if $TF_i$ matches to $\hat{h}^k$ in the first hop, $r^k$ is then used to output $\hat{h}^{k+1}$ which can match to $TF_j$ at the next hop. $\hat{h}^{k+1}$ is a joint representation of $\hat{x}$ and the current matched prototypes, represented by $r^k$, and the LSTM fine-tunes $w^k$ in order to find TF binding combinations.

The final output $\hat{y} \in \mathbb{R}^{|TFs|}$ is computed from a concatenation of the final hidden state and read vectors $[h^K; r^K]$ after the $K^{th}$ hop using a linear transform: $o = W([h^K; r^K])$. An element-wise sigmoid function is then applied to get a probability of binding for each TF: $\hat{y} = 1/(1 + e^{-o})$.

**Classification and Prototype Matching Loss Functions** To classify a sequence, we use a binary cross entropy loss between each label $y_i$ for $TF_i$, and the corresponding $TF_i$ output $\hat{y}_i$, as the classification loss, $\mathcal{L}_c$. We also introduce a prototype matching loss $\mathcal{L}_p$, which forces a prototype to correspond to a specific TF since prototypes are learned from random initializations. The prototype matching loss uses an L$_2$ loss between the true label $y_i$ for $TF_i$ and the final matching weight $w_i^K$ between updated sequence $h^K$ and prototype $p_i$. This loss forces a prototype to match to all of its positive binding sequences. The loss is computed from the final weights, $w^K$, after $K$ hops, which allows the LSTM to attend to certain TFs at different hops before the final loss, modeling the co-binding of TFs. $\lambda$ controls the amount that each prototype is mapped to a specific TF. $\lambda$=0 corresponds to random prototypes. The final loss $\mathcal{L}$ is a summation of the classification and prototype matching: $\mathcal{L} = -\mathcal{L}_c - \lambda \mathcal{L}_p$, where $\mathcal{L}_c = \sum_i^{|p|} (y_i \log \hat{y}_i + (1 - y_i) \log(1 - \hat{y}_i))$, and $\mathcal{L}_p = \sum_i^{|p|} (y_i - w_i^K)^2$.

Table 1: TFBS Prediction on 86 TFs. Shows PMN vs. two CNN baselines (single- and multi-label)

| Model | auROC | | | auPR | | | Recall at 50% FDR | | |
|---|---|---|---|---|---|---|---|---|---|
| | Mean | Std. | % Increase over single | Mean | Std. | %Increase over single | Mean | Std. | %Increase over single |
| **CNN (single-label)** | 0.820 | 0.072 | - | 0.263 | 0.123 | - | 0.224 | 0.198 | - |
| **CNN (multi-label)** | 0.831 | 0.055 | 1.37 | 0.257 | 0.113 | -2.52 | 0.215 | 0.186 | -4.00 |
| **PMN ($\lambda$=1), no LSTM** | 0.830 | 0.057 | 1.30 | 0.267 | 0.116 | 1.22 | 0.231 | 0.197 | 3.09 |
| **PMN ($\lambda$=1), softmax att** | 0.834 | 0.057 | 1.70 | **0.272** | 0.115 | **3.36** | **0.243** | 0.194 | **8.48** |
| **PMN ($\lambda$=0), sigmoid att** | 0.837 | 0.055 | 2.13 | 0.271 | 0.113 | 3.00 | 0.229 | 0.186 | 1.92 |
| **PMN ($\lambda$=0.5), sigmoid att** | 0.839 | 0.055 | 2.38 | **0.272** | 0.113 | **3.36** | 0.235 | 0.187 | 4.73 |
| **PMN ($\lambda$=1), sigmoid att** | **0.840** | 0.054 | **2.45** | 0.270 | 0.114 | 2.47 | 0.234 | 0.187 | 4.17 |

**Connecting to Previous Studies** Our method is motivated by the prototype-matching theory (Wallis et al., 2008), where instead of searching for exact features to match against, the model tests an unseen sample against a set of prototypes using a defined similarity metric to make a classification. Snell et al. (2017) introduced prototypical networks for zero and one-shot learning, which assumes that the data points belonging to a particular class cluster around a single prototype. Krotov & Hopfield (2016) show that pattern recognition is likely a combination of both feature-matching and prototype-matching. Our method models both the patterns of prototypes and the interactions among prototypes. In another line of work, Vinyals et al. (2015) and Vinyals et al. (2016) introduced the Matching Net for processing input sets where they use an orderless LSTM, similar to ours. Our model, however, learns the memory of the set and focuses on the large-scale multi-label classification tasks. The idea of refining the query based on a set of memory items (documents) has been explored similarly in Munkhdalai & Yu (2016) for QA (through a vanilla LSTM). To our knowledge, the PMN is the first work to use the memory-augmented NN model for multi-label prediction.

## 3 EXPERIMENTS AND RESULTS

**Dataset** We constructed our dataset from ChIP-seq experiments from Consortium et al. (2012). We then extracted the 200-length windows surrounding the peak locations for 86 transcription factors in the GM12878 cell line. Our training, validation, and test sets contain 1.44M, 330K, 300K sequences, respectively. Each sample has an average of $\sim 5$ positive labels (i.e., TFs binding).

**Model Variations** To test the PMN model on our TFBS dataset, we constructed 4 model variations:
**1. CNN:** As in Zhou & Troyanskaya (2015) and Lanchantin et al. (2016), we use a baseline 3-layer CNN model. This architecture is used as the encoder $f$ for all variations. We also implement a baseline single-task CNN which assumes no shared dependencies among TFs.
**2. PMN, no LSTM:** We use eq. 2-3, except that we replace $\hat{h}^k$ in eq. 3 with $\hat{x}$ since there is no LSTM. The output $o$ is then a concatenation of $r$ and $\hat{x}$. We still use the full prototype loss ($\lambda$=1).
**3. PMN, sigmoid att:** The full PMN model utilizes the CNN plus the combinationLSTM in eq. 1 over $K$ hops (We use $K = 5$ since each sample has on average 5 positive labels). We tested 3 variations of the prototype loss ($\lambda$=0, $\lambda$=0.5, $\lambda$=1), where $\lambda$=0 represents random prototypes.
**4. PMN, softmax att:** We replaced the sigmoid (eq. 3) from $k$=0 to $k$=$K$-1 with softmax (typically used in attention models), and then a sigmoid for the final multi-label output at step $K$.

**TFBS Classification Results** (Table 1) We evaluate our methods using area under ROC curve (auROC), area under the precision-recall curve (auPR), and recall at 50% false discovery rate. The joint CNN (multi-label) model outperformed the single label CNN models in auROC. The joint model's improvement over single-task was not significant (p-value < 0.05) based on a one-tailed pairwise t-test, presumably because the joint model finds joint motifs, but doesn't model interactions among TF labels. The PMN model outperformed both baseline CNN models in all 3 metrics, and was significant using a one-tailed pairwise t-test. We hypothesize that the combinationLSTM module accurately models co-binding better, leading to an increase in performance. We found that using the prototype loss, $\lambda > 0$, resulted in an improvement over random prototypes. We further plan to explore different similarity measures and prototype loss weights. Additionally, we plan to explore the interpretability of the prototypes as representations of classes, possibly in a generative manner. In conclusion, we found that our PMN method to update the input sequence based on output label dependencies can significantly improve mutli-label classification.

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
