# OpenReview forum: "Prototype Matching Networks for Large-Scale Multi-Label Classification"
_ICLR.cc/2018/Workshop — Reject_

### Official Review · AnonReviewer2 · 2018-03-08
**Little ML contribution, more suited for a genomics/computational biology conference.**

**Rating:** 4
**Confidence:** 4

**Review:**

The paper maps a pair (DNA sequence, transcription factor set) into a binary vector that indicate whether each transcription factor binds. The paper might have an application impact that I cannot judge but as far as machine learning is concerned, the contribution is not significant. I would recommend withdrawing from ICLR and submitting to a genomics/computational biology conference.

There are few non-standard choices which are not justified but I would not classify them as contribution.

a. The network rely on a fixed number of steps (5) in an LSTM, no new data is read at each step so essentially it is a deep network of depth 5 with parameter sharing across layers. Did you evaluate the impact of sharing as opposed to a deep network with sharing? Also it seems that the depth 5 is set a-priori (it matches the number of transcription factors in each set), however it might to be validated.

b. The loss function combines binary cross entropy (classical) and MSE (not classical for binary classification). This choice seems motivated by class imbalance although no motivation is given in the paper. Could you clarify? Also could you compare with classical solution for in-balance (ROC optimization see https://bmcfee.github.io/papers/mlr.pdf and reference therein or class weighting see https://papers.nips.cc/paper/2763-a-probabilistic-interpretation-of-svms-with-an-application-to-unbalanced-classification.pdf)?

Overall I feel this paper is more suited to a computational genomics conference.

---

### Official Review · AnonReviewer3 · 2018-03-09
**An interesting memory-augmented network for multilabel classification problems is presented. The experiments, however, are not fully convincing.**

**Rating:** 6
**Confidence:** 3

**Review:**

SUMMARY

In this work, the "prototype matching network" is proposed for modeling transcription factor binding mechanisms. Since there are two dominating classes of such binding mechanisms, the model proposed uses a specific encoder (based on a CNN) for modeling "motif-like" features, and a certain dependency model (based on an LSTM) to describe co-binding patterns. The purpose of this dependency model is to update an embedded sequence based on matches to prototypical transcription factors. This "combinationLSTM" uses the cosine similarity function, "squashed" through the logistic function to produce a similarity score.  The final model is applied to a  large-scale prediction problem involving
86 transcription factors.

EVALUATION
The paper is relatively easy to read. The motivation is clear, and the proposed memory-augmented network for a multilabel classification problem is certainly interesting. Although it shares some similarities with the model in [Munkhdalai & YU, 2016], this "combinationLSTM" can be considered as a interesting novel contribution. My main point of criticism, however, concerns the experimental evaluation. In particular, I am not fully convinced that the results in table 1 show indeed any significant improvement of the PMN model over the CNN baseline: first, the standard errors reported are much bigger than the differences. Second, the use of one-tailed pairwise t-tests is probably not appropriate here. How can you be certain that the consequences of missing an effect in the untested opposite direction is negligible? Are there still any significant differences in a two-tailed test remaining?

---

### Official Review · AnonReviewer1 · 2018-03-09
**Lack of strong evidence to show significant methodological improvement**

**Rating:** 4
**Confidence:** 2

**Review:**

The explanation of the methodology is challenging due to the short paper. It is difficult to understand how the approach is constructed. From what I can understand, the methodology seems interesting, using CNNs as has been done before to provide feature extraction but then the LSTM to model dependencies in the labels. However, the main problem of the paper is that the results do not appear to suggest that the PNM method gives considerably better performance than the baseline models. Although the authors cite that the improvement is statistically significant, the absolute differences in the benchmarking metrics are not very large. Furthermore, results depend on the value of \lambda in the loss function and whether PNM is based on an LSTM or not -- there is no variant of the model which consistently gives the best results.

---

### Decision · Program_Chairs · 2018-03-20
**ICLR 2018 Workshop Acceptance Decision**

**Decision:**

Reject

**Comment:**

Based on the reviews, this paper has not been accepted for presentation at the ICLR workshop. However, the conversation and updates can continue to appear here on OpenReview.